# Nonlinear Image Registration and Pixel Classification Pipeline for the Study of Tumor Heterogeneity Maps

**DOI:** 10.3390/e22090946

**Published:** 2020-08-28

**Authors:** Laura Nicolás-Sáenz, Sara Guerrero-Aspizua, Javier Pascau, Arrate Muñoz-Barrutia

**Affiliations:** 1Departamento de Bioingenieria e Ingenieria Aeroespacial, Universidad Carlos III de Madrid, 28911 Leganes, Spain; lnicolas@ing.uc3m.es (L.N.-S.); sguerrer@ing.uc3m.es (S.G.-A.); jpascau@ing.uc3m.es (J.P.); 2Centre for Biomedical Network Research on Rare Diseases (CIBERER), U714, 28029 Madrid, Spain; 3Hospital Fundación Jiménez Díaz e Instituto de Investigación FJD, 28040 Madrid, Spain; 4Epithelial Biomedicine Division, CIEMAT, 28040 Madrid, Spain; 5Instituto de Investigación Sanitaria Gregorio Marañon, 28007 Madrid, Spain

**Keywords:** computational pathology, image registration, antigen segmentation, cancer

## Abstract

We present a novel method to assess the variations in protein expression and spatial heterogeneity of tumor biopsies with application in computational pathology. This was done using different antigen stains for each tissue section and proceeding with a complex image registration followed by a final step of color segmentation to detect the exact location of the proteins of interest. For proper assessment, the registration needs to be highly accurate for the careful study of the antigen patterns. However, accurate registration of histopathological images comes with three main problems: the high amount of artifacts due to the complex biopsy preparation, the size of the images, and the complexity of the local morphology. Our method manages to achieve an accurate registration of the tissue cuts and segmentation of the positive antigen areas.

## 1. Introduction

Cancer diagnosis is determined by broad image and pathology studies. In these pathology studies, insight into molecular and cellular interactions, growth, and organization is obtained through careful analysis and observation of stained histological sections. These studies are complex, long, subjective, and prone to error. However, misdiagnosis could have fatal consequences, as would taking too long to determine the appropriate diagnosis. Therefore, this process needs to be automatized, to make it fast, accurate, robust, and objective.

To show the importance of this work, an explanation of the current diagnostic techniques follows.

When a biopsy is taken of a suspicious mass, it is sectioned into thin slices and stained with Hematoxylin and Eosin (H&E). The majority of the diagnoses are performed based on the morphological study of the tumor from H&E stained tissue. However, in many cases, to complement a difficult differential diagnosis, or to better understand the biology of the tumors, the sections may be stained with different antigens. Each antigen marks a function relevant for the diagnosis.

After staining the biopsy sections with different markers, the pathologist is able to extract related information regarding its structure and biology. In many cases, the tumor’s intrinsic heterogeneity is important because it is now understood that malignant tissue can encompass subpopulations of cells with specific genomic alterations within the same tumor [1]. Thus, the developing tumor cells may express distinct molecular signatures that respond deferentially to different therapeutic options. Moreover, this heterogeneity also affects the acquisition process, because the inter-tumor variability can be very large, it is important to take sufficient samples so that no information is missed. Therefore, in order to analyze the tumor heterogeneity, it is necessary to take several samples that span the whole of the mass and stain them with different dyes to study all the distinct morphological characteristics and functions. Once the sample preparation is ready, the mounted slices have to be scanned and digitized. With all the histopathological data along with the image tests the pathologist will provide a more complete diagnosis, including staging and grading of this tumor [2].

This calls for an efficient new way of conveniently visualizing such huge amounts of data that saves time and effort and possibly provides more information than the system in place. Our proposal is a tumor heterogeneity map.

In this paper, we propose an entirely automatic, non-supervised method for the creation of tumor spatial heterogeneity maps. These maps would provide a novel way of histopathological data visualization, easily representing how the different biomarkers are distributed over the tissue cuts. This approach would allow the clinician to efficiently discern which areas are more proliferative, which more angiogenic, and what are the relations between them. Heterogeneity maps will aid in understanding the tumor’s microenvironment and will enable the scoring of the tumor using the automatically detected neoplastic areas.

The creation of these maps involves the use of state-of-the-art image registration algorithms combined with a unique approach for antigen localization and segmentation as seen in Figure 1. Code is available at [3].

### 1.1. Histological Sections Registration

Stacking high-resolution 2D images of histopathology can provide much richer information than simply studying them individually because it can lead to new understanding of certain features and colocalization of biomarkers. This stacking, however, requires the registration of the individual biopsy slices.

Image registration is defined as the process by which two or more images of a set are aligned together and correlated in order to associate information common to these images.

There are three main problems when it comes to histopathology image registration. The first problem is that there is not a conserved shape for biopsies: Each one is unique and depends on multiple factors such as the tumor, the patient, the staining, and the conditions under which it was extracted. Moreover, there are no reference points that can be used to guide the alignment process and the order of the slices is hardly ever known. Therefore, it is necessary to select one of the slices as the *template* or *fixed* image and match the rest of the *moving* images to it.

The second problem is the size of the files. These images are acquired into very high resolution and thus are very large in matrix size. This translates in problems with storing the files as well as the high computer power needed to work with them. Due to this, it is common to work with Whole Slide Images (WSI), which are high-resolution digital files that can be efficiently stored and accessed as they include different resolutions of the same image in a pyramidal fashion.

The third problem stems from the difficult acquisition process. Because the slices are cut so thinly and they are so fragile, sample preparation introduces misalignments and artifacts like rips, folds, and tissue compression and stretching. In the worst cases, there are severe stain variations between slices and torn or even missing tissue. Moreover, there is no standardized procedure for digitization, and so the slices are hardly ever positioned all in the same way, which introduces displacements and rotations between slices of the same biopsies. These issues add more complexity to the image registration process.

Because these artifacts affect the biopsy at a morphological level, rigid registration is not the correct approach. Histopathology image registration needs the use of local, non-rigid registration algorithms that consider the small, unpredictable morphological changes between biopsies caused by the acquisition process. These issues explain why image registration has not yet been widely applied to computational pathology and why it is so important to develop a robust method that can tackle these hurdles.

### 1.2. Antigen Localization and Segmentation

The localization and segmentation of positive antigen areas in digitized histopathological biopsies is still one of the biggest challenges in computational biology. This is mainly due to the completely different color appearances that these biopsies can take. Due to the lack of standardized procedures for staining, each technician or technique variances can introduce considerable differences in the final image. Developing a system to perform this segmentation has thus been a persistent challenge in computational pathology, as creating a method that can perform robustly over different stains, tumors, conditions, and techniques is yet to be found [4]. Moreover, immunohistochemical (IHC) staining adds extra difficulty to the problem due to the colors the stains have. IHC stained biopsies show a mixture of two very similar colors: purple (from the counterstaining) and brown (from the peroxidase)—both a mixture of red and blue—introducing additional difficulty for automatic systems to segment the stains.

The method we used (described in Section 3.4) was previously tested and applied to the segmentation of several positive antigen stains on colon adenocarcinoma samples. Our approach will use this innovative color segmentation algorithm on previously registered biopsies in order to obtain the areas positive for the antigen in each of the already registered slices. The stacking of these areas will thus give rise to the tumor heterogeneity maps, from which to study the tumor microenvironment

## 2. State-Of-The-Art

All image registration algorithms have a set of common elements. They attempt to create a geometric transformation that efficiently maps a moving image onto a fixed image employing the optimization of an error metric [5] as reflected in Figure 2. This general pipeline is constructed in order to get over the main hurdles of pathology image segmentation: the immense size of the images, the thin sectioning which causes artifacts and local errors, and the variability between samples.

However, each algorithm proceeds differently. In 1998, Mainz et al. [6] described the classification criteria for image segmentation that was perfected in 2016 by Viergever et al. [7]. This classification is as follows. (1) Dimensionality: depending on the dimensions of the set of images. (2) Nature of the transformation: it can be linear when the internal structure has not suffered important distortions: rigid, affine, and projective transformations, or nonlinear, which is perfect for local deformations on which linear transformation fails and be based on physical models (Demons [8,9]) or on basis functions (B-splines and radial basis function [10]). (3) Domain of the transformation, i.e., whether the transformation is local or global. (4) Degree of interaction: If the algorithm needs annotations and supervision (semi-automatic) or not (automatic). (5) Optimization procedure: which may be continuous or discrete.

The similarity metric greatly varies between algorithms, depending on what is optimized. There are intensity-based algorithms that attempt to maximize an information theory metric such as mutual information (Elastix [11], Advanced normalization [12]). There are also feature-based registration methods, which extract descriptive features using mostly Scale-Invariant Feature Transform (SIFT) or Speeded-Up Robust Features (SURF) and estimate an optimal transformation using RANdom SAmple and Consensus (RANSAC) (feature-based registration). Feature- and intensity-based methods mix both previous approaches by using the feature method for the estimation of a rough initial alignment and then optimize the alignment using intensity metrics for similarity (feature-based+Elastix [13]). Landmark-based algorithms try to minimize the distance between two sets of landmarks. They are easily recognized and conserved features that are manually selected and placed on the images by experts. Finally, segmentation-based methods align binary structures (SegReg [14]). Many of these registration methods—mainly feature- and landmark-based—are supervised algorithms, due to their dependence on manually annotated regions or landmarks [15,16,17].

Another important issue with registration methods is the type of images they attempt to align. Considering this, the process may be multimodal, if they are registering images from different sources (CT and MRI) or monomodal, if the source of both images is the same. Nevertheless, there is disagreement on what can be considered monomodal, as many argue that the registration of differently stained histological sections can be considered a multimodal problem due to the high variability in appearance between the stained images [5].

As it has been shown, image registration has been thoroughly researched in computational pathology and yet remains an unresolved topic. Computational pathology analysis of consecutive differently stained cuts is not yet routinely established due to a lack of robustness of the proposed deformation methods.

Concerning color segmentation, many advances have been achieved for H&E staining [18,19], but there are few methods that can robustly segment the positive antigen stain of immunohistochemical biopsies.

For unsupervised approaches, the most used method was proposed by Ruifrok et al. [20], which is based on a deconvolution matrix and is commonly referred to as color deconvolution algorithm. This algorithm assumes a linear relationship between stain concentration and absorbance (Beer’s Law), which is not true for IHC samples. However, most researchers disregard this problem and use a modification of Ruifrok’s original deconvolution matrix with acceptable success. Versions of this matrix are available for different normalized staining in public software tools such as Cell Profiler [15,21].

The supervised approaches rely on machine learning or deep learning algorithms that, from a set of annotated biopsies, are able to learn a general segmentation [22,23]. However, due to the high variability between samples, the lack of standardized procedures, and the particular difficulty of IHC staining segmentation due to their brown-purple appearance, these methods have not yet been able to produce robust and generic solutions.

## 3. Materials and Methods

In this section, we present the images we worked with and the software and machines we used, as well as the algorithms we developed. The pipeline we follow is represented in Figure 1, Figure 2, Figure 3 and Figure 4.

### 3.1. Image Data

For this work, we used publicly available data from the Automatic Non-rigid Histological Image Registration (ANHIR) challenge 2019 [5,24,25]. This challenge was set up to compare the performance of different registration algorithms on a varied set of microscopy histology images. They provided eight different datasets with images from various sources, all of them acquired as high-resolution whole slide images. Alongside the images, the organizers of the challenge provided a series of landmark files. These landmarks point to important structures in the tissue and were manually annotated in each image by four different professionals with a difference of a 0.05% average distance of the image diagonal between annotators. The landmarks were offered as csv files with 86 annotated points per image. The landmark files were used as a tool for the assessment of the registration results. Because this project is centered in the study of tumor heterogeneity, we did not analyze all the datasets, but rather just the ones that included carcinoma biopsies: The Gastric Adenocarcinoma and the Lung Lobes datasets.

The lung lobes set was comprised of mice unstained adjacent 3 μm formalin-fixed paraffin-embedded sections stained with Hematoxylin and Eosin (H&E) or by immunohistochemistry with a specific antibody for CD31, proSPC, CC10, or Ki67. These sections were obtained from either adenoma or adenocarcinomas using a Zeiss Axio Imager M1 microscope (Carl Zeiss, Jena, Germany) equipped with a dry EC Plan-Neofluar objective (NA = 0.30, magnification 10×, pixel size 1.274 μm/pixel). From this dataset, we obtained 4 mice with 5 Whole-Slide Image (WSI) tissue cuts and 5 landmark files each, which summed up to a total of 20 images and 20 landmark files. The stains employed for the immunohistochemistry studies of this dataset are used for demonstrating the presence of endothelial cells in histological tissue sections for the evaluation of the degree of tumor angiogenesis (CD31,proSPC), inflammation (CC10), and cellular proliferation (ki67).

For the gastric set, the organizers uploaded biological material from patients with a histologically verified diagnosis (gastric adenocarcinoma). This material was used for immunophenotyping in order to study the cellular composition of the tumor tissue infiltrate. The study of the cellular composition of the tumor tissue infiltrate was performed by obtaining 4 μm adjacent slices and using immunohistochemical staining on 4 markers of the cluster of differentiation (CD) family. This family of biomarkers is used in protocols for cell immunophenotyping [26]. The used markers were CD1a (clone O10), CD4 (clone 4B12), CD8 (clone C8/144B), and CD68 (clone PG-M1). CD1a is a membrane stain used for Langerhans cells either inflammatory or neoplastic. CD4 stains cytoplasm with accentuation in the membrane. It is used to mark T cells. CD8 is a membrane stain that is also used to stain T cells. CD68 is a nuclear stain that marks histiocytes. Deparaffinization and antigen recovery were performed by using Thermo Dewax and HIER Bufer L, pH6 buffer. The preparations were studied with a Leica DM LB2 microscope by two independent researchers. The digitization was performed with a Leica Biosystems Aperio AT2 scanner at 40x with a resolution of 0.2528 (μm/pixel). From the gastric set we had 9 patients with each 4 Whole-Slide Image (WSI) tissue cuts and 2 landmark files, which summed up to a total of 36 images and 18 landmark files.

All in all, we had 56 Whole Slide Images and 38 landmark files on which to test our registration algorithm.

Because the computational cost (time and memory) of using the full-resolution images for the registration was unacceptable, we used 5% resolution for the gastric set and 10% resolution for the lung lobes set. This meant having an average size of 5 k × 3 k pixels and a resolution of 3 (μm/pixel) for the gastric set and a size of 2 k × 1.5 k with a resolution of 7 (μm/pixel) for the lung datasets. Nevertheless, the original size of the files was used for the color deconvolution algorithm.

Despite careful sample preparation, artifacts such as rips, loss of tissue, and folds were observed in all images, which further complicated the process of registration. Examples of these artifacts can be seen in Figure 3.

### 3.2. Computer

The software was developed using was a cluster with CPU Intel Xeon 4110 in 2 sockets, 16 cores, 32 threads, 2.1 GHz, 64 GB RAM, and 2400 MHz DDR4. The server was a SKY-6000 Advantech. All the algorithms and their respective evaluations were carried out using MATLAB 2019.

### 3.3. Image Registration

For the biopsy registration, we need the best transformation $T:\left(x,y\right)\to \left(x^{\prime },y^{\prime }\right)$ that can map the moving image I(x,y) into its corresponding points in the fixed image $I(x^{\prime },y^{\prime })$. For this, we developed a transformation *T* that consists of a robust pre-alignment and a global and a local transformation. Because the images were all of the same modality, we decided to use intensity-based registration methods. The combined transformation *T* can be expressed as $T=T_{Align}+T_{Global}+T_{Local}$ as seen in Figure 4.

The alignment and global motion are used to describe the overall motion of the biopsy and appropriately register the biopsy global shape. Therefore, it is applied not to the biopsy sections, but rather to a rough binary mask of each of them, which takes into account neither the possible rips, holes, or folds of the tissue nor its local details. The local motion involves the alignment of the internal structure of the biopsy, and thus it was computed using the globally-transformed RGB images. This local motion model is needed because the global motion does not take into account the internal structure of the biopsy. The nature of the local deformation of a biopsy is unique, individual, and varies depending on multiple factors (e.g., sample preparation and image acquisition) and thus cannot be modeled by rigid registration algorithms nor can it be tackled with large scale non-rigid algorithms such as the ones used for Global registration.

The whole registration process is described in detail in the next paragraphs.

#### 3.3.1. Pre-alignment ($T_{Align}$)

As previously explained, it is difficult to preserve the shape and position of the samples during digitization, and consequently some of the images were rotated and displaced, in addition to having folds and rips of the tissue. Due to this, the images had to be properly aligned before complex registration could be performed. This is done as a separate step instead of being included in an Affine transformation as the movement limitations were different. For the alignment, the shape of the biopsy section was not modified, but merely displaced and rotated into its best position. Because of this, we had no movement restrains for this transformation.

For this alignment, the centers of mass of the moving and fixed images are detected, and the vector pointing from one center of mass to the other is used to perform the initial translation. Once both images are aligned, we compute the optimal rotation of the moving image binary mask by maximizing the cross-correlation value over 10-degree increments from 0${}^{\circ }$ to 355${}^{\circ }$. Therefore, the final pre-alignment transformation is $T_{Align}=T_{Displacement}+T_{Rotation}.$

#### 3.3.2. Global Model ($T_{Global}$)

The global motion is used to describe the overall motion of the biopsy. This model is primarily used to appropriately register the global shape of the biopsy. Therefore, it is applied to a rough binary mask of each section instead of the section image itself.

Because this initial transform only matches the global shape of the biopsy, we use affine registration, and then a mild non-rigid registration for refinement on top of the displacement and rotation of $T_{Align}$. As the affine transformation modifies the shape and local structures of the biopsy, mild deformation limitations must be enforced to avoid drastically deforming the internal structures.

Once the moving image binary mask has been subjected to $T_{Align}$, an affine, intensity-based transformation is calculated between the fixed and the moving binary masks, using 3 Gaussian multi-resolution pyramid levels for refinement.

The optimization is computed over a series of iterations using regular step gradient descent with mean square error as the error metric. The initial parameters are shown in Table 1.

After the affine transformation, we applied a postprocessing step that roughly fixed the final edge mismatch using a diffeomorphic demons nonrigid transformation [27], developed in [8]. For this algorithm, a mesh of demons is created where the “demons” are control points that can either be located throughout the whole image (one demon per voxel) or only over the contours. In this case, we used the mesh over the whole image. The algorithm is based on diffusing models of attraction. For each iteration, a deformation of each demon is caused by a dragging demon force as given in Equation (Equation 1), which is calculated based on diffusion equations:(1)$$v=\frac{\left(I_{m}-I_{f}\right)\vec{\nabla }I_{f}}{{\left(\vec{\nabla }I_{f}\right)}^{2}+{\left(I_{m}-I_{f}\right)}^{2}}$$
where $I_{m}$ and $I_{f}$ are the intensities of the moving and fixed images, respectively, and $\vec{\nabla }I_{f}$ is the gradient of the fixed image. In order to get a smooth displacement field, Gaussian field smoothing is applied to the whole field after every iteration.

Diffeomorphic demons present the added advantage of symmetric demon forces, which yield faster convergence, and thus make this transformation a one-to-one smooth, continuous mapping with invertible derivatives.

We used the Matlab implementation of diffeomorphic demons non-rigid registration with 3 multi-resolution pyramid levels, 100 iterations, squared sum of differences (SSD) as similarity measure, and Gaussian field smoothing for regularization on top of 1.0 smoothing applied at each iteration.

The transformation $T_{Global}$ is applied to the original moving image: $T_{Global}=T_{Affine}+T_{Demons}$.

#### 3.3.3. Local Model ($T_{Local}$)

In order to match the local deformations between biopsies, we used a free form deformation (FFD) based on B-splines following the work of Rueckert et al. [28]. In order to take into account the interior details of the slide, both $T_{Align}$ and $T_{Global}$ are applied to the original image. To improve computational efficiency, the biopsy section was segmented from the background of the slide.

FFD works by the manipulation of a mesh of control points over the whole image. This mesh is optimized and produces a smooth continuous transformation that is later applied to the moving image. For the definition of the FFD, we describe the domain of the image plane as $\Omega =\left\{(x,y)|0\leq x<X,0\leq y<Y\right\}$ and $\phi_{i,j}$ as the $n_{x}\;\times n_{y}$ mesh of control points with uniform spacing μ.

The FFD is thus defined as the 2D tensor product of the B-splines:(2)$$T_{local}\left(x,y\right)=\sum_{l=0}^{3}\sum_{m=0}^{3}B_{l}\left(u\right)B_{m}\left(v\right)\;\phi_{i+l,j+m\;}$$
(3)$$B_{o}\left(u\right)=\frac{{\left(1-u\right)}^{3}}{6};B_{1}\left(u\right)=\frac{\left(3u^{3}-6u^{2}+4\right)}{6};B_{2}\left(u\right)=\frac{\left(-3u^{3}+3u^{2}+3u+1\right)}{6};B_{3}\left(u\right)=\frac{u^{3}}{6}$$
where $i=\left[x/n_{x}\right]-1;j=\left[y/n_{y}\right]-1;u=x/n_{x}-\left[x/n_{x}\right];v=y/n_{y}-\left[y/n_{y}\right]$ and $B_{l}$ represents the lth basis function of the B-spline.

Equation (Equation 2) shows that the B-spline FFDs are controlled locally due to limited support of the basis functions. Therefore, the movement of a control point affects only its neighboring points in a 4 μ× 4 μ area [28], which represents an advantage over thin-plate splines and elastic-body splines because it makes the computation more efficient.

The control points $\phi_{i,j}$ are the parameters of the B-spline FFD. The degree to which the mesh of control points can be deformed and modeled depends on its resolution, which is inversely proportional to its uniform spacing [29]. For computational efficiency, we used a uniform spacing of [50,50] pixels.

While the FFD is slower than the diffeomorphic demons, we chose it for the local model because the final transformation field corresponds better to the deformations found in clinical cases [30].

Cost Function 

To compare the two images for registration, a similarity criterion is needed. This criterion measures the extent of the alignment between the fixed and the moving images. The logarithmic difference metric defined by Equation (Equation 4) was the similarity function we used for this purpose.
(4)$$C_{\mathrm{sim}}=\log\left(abs\left(I_{M}-I_{F}\right)\right)$$

Optimization 

The goal of B-spline FFD is to find the best deformation of the control points mesh so that the moving and fixed images are as aligned as possible. For this, the best local transformation parameters ϕ have to be calculated. The cost function describes this with two terms: the first ($C_{\mathrm{sim}}$) is the cost of the image similarity, while the second ($C_{\mathrm{smooth}}$) is related to the smoothness of the final transformation:(5)$$C\left(\phi \right)=-C_{\mathrm{sim}}\left(I_{F},TI_{M}\right)+\lambda C_{\mathrm{smooth}}\left(T\right)$$
where lambda λ is the regularization or penalty function—“thin sheet of metal bending energy” [31], which measures the smoothness trade-off and it is critical for regularization of dense control meshes. Therefore, the smoothness parameter has to be chosen taking into account the importance of model flexibility versus computational complexity. In our case, it was empirically tested that λ=0.01 was the best value for this parameter.

For computational efficiency, the optimization is done in subsequent steps. In the first step, the control points ϕ are optimized as a function of the cost function. This optimization is done through iterative steepest descent using a quasi-Newton optimizer for nonlinear least squares problems implementing a trust-reflective algorithm [32]. When the algorithm finds a local optimum (||ΔC||≤ϵ for some small positive value) of the cost function, or when it reaches the iteration limit, it stops. Given our computational power, we used a limit of 70 iterations and a stop criterion of ϵ = $1.0\times 10^{-6}$.

#### 3.3.4. Evaluation of Registration Performance

It is important to understand that pixel perfection can never be achieved with pathology registration because structures change and evolve between sections. Nevertheless, visual assessment is not enough to measure the efficiency of a registration pipeline. Thus, the proposed algorithm was evaluated using the two most common numerical metrics: the Landmark Distance Evolution and the Registration Confidence Map: Overlap measurements.

(1).Landmark Distance evolution

The data used for this project came from the ANHIR image registration challenge and it included landmarks for half of the images (two landmark files per patient). The proposed metric of this challenge was the relative Target Registration Error (rTRE), which is the euclidean distance between the coordinates of the landmark points ($x_{l}^{i},x_{l}^{j}$) of the fixed and moving images normalized by the length of the diagonal [5]:(6)$${\mathrm{rTRE}}_{l}^{ij}=\frac{{∥x_{l}^{i}-x_{l}^{j}∥}_{2}}{d_{j}}$$
where $l\in L^{i}$ and $L^{i}$ is the set of landmarks.

With this metric they then defined a series of additional metrics with the purpose of taking into account the outliers and the differences in difficulties between datasets and images.

These additional metrics are computed as follows.

Average median rTRE:(7)$$\;\mathrm{AMrTRE}\;\left(m\right)={\mathrm{mean}}_{(i,j)\in T}\mu^{i,j}\left(m\right)$$

Median of median rTRE (MMrTRE):(8)$$\;\mathrm{MMrTRE}\;\left(m\right)={\mathrm{median}}_{(i,j)\in T}\mu^{i,j}\left(m\right)$$
where
(9)$$\mu^{i,j}\left(m\right)={\mathrm{median}}_{l\in L^{i}}r{\mathrm{TRE}}_{l}^{i,j}\left(m\right)$$

Average maximum rTRE (AMxrTRE):(10)$$\mathrm{AMxr}\mathrm{TRE}\left(m\right)={\mathrm{mean}}_{(i,j)\in T}\max_{l\in L^{i}}{\mathrm{rTRE}}_{l}^{i,j}\left(m\right)$$
where T is the set of all the image pairs.

Furthermore, they defined a way of calculating the robustness of the method as the relative number of successfully registered landmarks.
(11)$$R^{i,j}\left(m\right)=\frac{\left|K^{i,j}\right|}{\left|L^{i}\right|}$$

(2).Registration confidence map: overlap measurements

Landmark distance evolution can only measure registration at the points where the landmarks are, but it does not take into account the global aspect of the transformation. Therefore, we also evaluated our results by computing the percentage of correlation between registered biopsies [15]. This was estimated as the overall percentage of overlap (pure intersection) of the binary masks of the tissue sections for each patient before and after registration. In order to calculate it, an exhaustive binary mask is computed for each section after global registration such that it takes into account all of the little details, tears, holes, etc. of each slide. The stacking of these binary masks is what forms the registration confidence map. Analytically, it is measured by adding up these binary masks so that, as 4 sections per patient were evaluated in the Gastric dataset and 5 sections per patient in the Lung Lobes dataset, the measured overlap ranges from 1 (no overlap) to 4 (overlap of all slides) in the case of the Gastric dataset and from 1 (no overlap) to 5 (overlap of all slides) in the case of the Lung Lobes dataset.

### 3.4. Antigen Localization and Pixel Classification

For the color deconvolution algorithm, a working pipeline based on a robust computational method for the segmentation and scoring of immunohistochemical samples capable of analyzing tumor biopsies fast and effectively. The segmentation method is based on the bisection of the red-blue color joint histogram of the images through their diagonal. This process is shown in Figure 5. This method is very simple and its efficacy was tested using immunohistochemical biopsies of colorectal cancer of three different patients stained with eight different antigens.

This method was validated for the new data using a ground truth created with the WEKA trainable segmentation algorithm. The WEKA tool is a plug-in of Image J that combines machine learning algorithms with selected image features to produce pixel-based segmentations [33]. It computes a segmentation based on an initial classifier, giving as final output two binary images—one for biomarker and another for nuclei—by analyzing the probability that each pixel belongs to a class. However, this segmentation tool needs manual initialization, with each stain needing to be initialized separately. Because of the size of the images, it was more efficient and robust to “break up” the biopsies into 512 pixels × 512 pixels tiles and perform the segmentation over each section individually.

During training, two classes were contemplated: biomarker (antigen) and background. With these classes, we performed manual labeling of 4 tiles per stain for each of the patients with ten labels per tile. Then, the program was trained with these labels using multilayer perceptron classifier [34]. Once trained, it segmented and classified the designated classes in the rest of the tiles of a particular biomarker staining. The two-level stack was then used to create the binary masks for each class.

The color deconvolution algorithm was applied to the already registered tissue cuts.

#### Evaluation of Segmentation Performance

The segmentation performance was evaluated comparing the binary masks generated by our algorithm with the ground truth generated with WEKA. The metrics used were the Dice similarity coefficient and Hausdorff distance.

**Dice similarity coefficient**: spatial overlap index between two sets of binary segmentation results. Its value ranges from 0, indicating no spatial overlap, to 1, indicating complete overlap [35].
(12)$$\mathrm{Dice}\left(A_{B},W_{B}\right)=\frac{2\left(A_{B}\cap W_{B}\right)}{\left|A_{B}\right|+\left|W_{B}\right|}$$
where $A_{B}$ and $W_{B}$ are the binary masks for the algorithm and WEKA images, respectively.

**Hausdorff distance**: maximum distance from a set $F_{B}$ to the nearest point in the other set [36]:(13)$$h\left(A_{B},W_{B}\right)=\max\left\{\min\left\{d_{e}\left(a,b\right)\right\}\right\}$$
where $d_{e}$ is the Euclidean distance.

## 4. Results and Discussion

### 4.1. Landmark Registration

The evolution of the registration error in the data after each step of the process is shown in the box plots in Figure 6. The median error is effectively reduced after each step in all cases.

The full results are available in Table 2 (Gastric) and Table 3 (Lung Lobes), where we compare our algorithm to several well-known and freely available registration methods, which were tested in [5]: bUnwarpJ, Register virtual stack slices (RVSS), NiftyReg, Elastix, advanced normalization tools (ANTs), and DROP. This shows that our method can compete with state-of-the-art methods, outperforming most of them. The performance of our proposed pipeline is demonstrated with Gastric 3 sample in Figure 7 and Figure 8.

### 4.2. Registration Confidence Map

#### 4.2.1. Global Motion Model

We define the registration confidence map as a measure of the overlap between the registered sections. The goal of a successful registration would thus be maximizing the percentage of the map that has complete overlap. However, it is important to notice that a 100% overlap would not be realistically achievable. Perfect alignment for the whole biopsy is not something to aspire to, as the changing shape and morphology of the sections throughout the biopsy must be taken into consideration. Table 4 and Table 5 show the results for the overlap possibilities. The time required to run the Global Motion Model averaged to five minutes per pair.

Figure 9 shows the overlap for all sections of the patient in Figure 7.

#### 4.2.2. Local Motion Model

For the Local Motion Model we followed the same procedure as with the Global Motion Model. The application of the local transformation improved overlap percentage in all cases, which can be seen in Table 4 and Table 5. The poorer results for the Lung Lobes dataset are explained by the nature of the images (seen in Figure 3), which are mostly mesh-like tissue with few solid lesions. The time required to run the Local Motion Model averaged to two hours per pair.

#### 4.2.3. Pixel Classification

The final results for the Dice similarity coefficient have a mean of 0.97 and a standard deviation of 0.0352. The final results for the Hausdorff distance have a mean of 10.96μm and a standard deviation of 3.97μm. The threshold of acceptance for this would be the average diameter of a cell, which is of 25.48μm. The elapsed time for computation of 100 tiles was 20 s for our algorithm and 20 min for the WEKA tool. The performance of both algorithms can be visually assessed in Figure 10, where it can be seen that, while WEKA performs marginally better, the improvement is not enough to justify the extra time and effort needed for the training and testing of the model.

These results show that our color segmentation method is highly robust, efficient, and entirely non-supervised.

### 4.3. Final Heterogeneity Map

The final Heterogeneity Map was created using the registered tissue cuts. Each transformed section was subjected to our color segmentation algorithm, which gives as output a binary mask of the areas positive for the antigen. These binary masks were stacked together over the fixed image selected for that patient. This creates a map of the antigen distribution over the general shape of the biopsy. In this case, we created two heterogeneity maps: one with the Lung Lobes set, and another one with the Gastric set. The maps, which can be seen in Figure 11 and Figure 12, coincide with the antigen distribution observed in each of the tissue cuts. From the map in Figure 11, we can observe that the endothelial cells (CD31, in green), are located around the lesion, but not within, which is mostly dominated by the pneumocytes forming the alveolar–capillary barrier (ProSPC, in lilac). The dividing nuclei (Ki67, in red) are also located within the lesion and can be found by themselves or in the same areas as ProSPC (in blue). Clara cells (Cc10, in orange) are only found outside of the lesions and mostly lining the small airways.

From the map in Figure 12, we can observe that the helper T cells (CD4, in orange) are distributed over most of the biopsy (light blue with CD1a, maroon with CD68a, navy blue with CD68a and CD1a, pink with CD68a, and CD8a with green) except for the center, which lacks any stain. Lipid antigen-presenting molecules (CD1), Histiocytes (CD68a), and Cytotoxic T-cells (CD8a) are mostly present along the upper edges of the biopsy, with Histiocytes being the most widespread of the three.

In future work, we would like to apply this pipeline to biopsies stained with more diverse markers so as to study to colocalization of these and be able to generate comprehensive heterogeneity maps that showed the colocalization of different biomarkers, as well as separate heatmaps for each different cell function.

## 5. Conclusions

We have presented a method to create tumor heterogeneity maps using a novel registration technique combined with a novel segmentation technique, both of which are completely automatic. The advantages of our proposed pipeline are its robustness and, most importantly, its completely automatic and non-supervised character. Compared to other segmentation and registration algorithms, our proposal yields similar registration results and improved segmentation results without the need of manual annotations nor training. The limitations of this pipeline are purely data-driven, as it is entirely automatic and non-supervised. Thus, the generation of the heterogeneity maps may fail if the sections are poorly stained and show no signal or if the slides have suffered extensive damage during sample preparation. However, these limitations are to be expected, as we still do not have algorithms that can recover missing data without failure. The created tumor heterogeneity maps provide a helpful visual representation of the intra-tumor environment and heterogeneity of the tissue. By showing the exact location of the stains within the tissue and how different dyes may correlate, this tool presents a new approach to the study of the tumor microenvironment. Moreover, once real-world testing in clinical settings is carried out and the maps prove to correlate with human-based conclusions, these maps would serve as an aid in visual assessment of tumor biopsies. Of course this is not aimed at substituting pathologists. The goal would be to hand over this tool to the clinicians, who then could use it for research in tumor heterogeneity, for the identification of interesting areas and slices, or even as a tool to speed up the process of analyzing biopsies for a cancer diagnosis. This last item could mean reduced diagnostic times with more accurate decisions; the use of these maps would be highly beneficial for speeding up and relieving the clogged system by making the whole process more efficient.

These maps can also serve as automatic, non-supervised computer-generated input for the training deep convolutional neural networks (DCNNs). Nowadays, these networks rely on manual annotations for training, which means they are dependent on the availability of pathologists. With our maps, tumoral areas and normal tissue can instead be automatically segmented and DCNNs can be trained to detect the corresponding structural information in clinical settings.

## Figures and Tables

**Figure 1 entropy-22-00946-f001:**
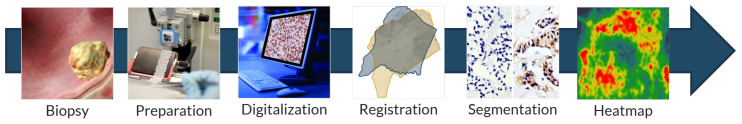
Visual illustration of project’s pipeline, from the clinical setting to computational pathology. The process starts with the extraction of the biopsy, which is prepared and digitized. After this, the digitized tissue cuts are registered and their positive antigen segmented to create final heatmaps.

**Figure 2 entropy-22-00946-f002:**
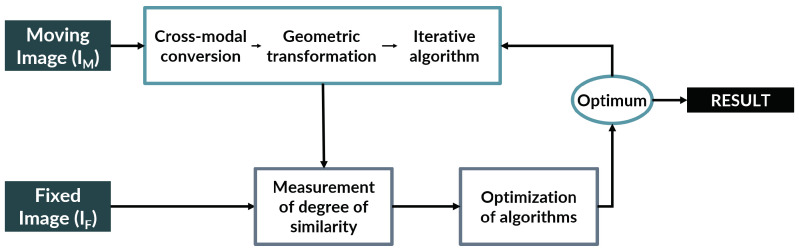
General pipeline of registration algorithms. This figure illustrates how general registration algorithms work: having a reference (Fixed Image) and an object (Moving Image), the Moving Image is subjected to a series of operations to attempt to map it to the Fixed Image. The quality of this mapping is evaluated by measuring the degree of similarity between the two images. The registration algorithm is optimized until the mapping is deemed optimum. This optimum mapping is decided with a cost function that will either reach an acceptable value or stop after a maximum number of iterations. The final result of the pipeline is a modified Moving Image that resembles the Fixed Image to the maximum possible degree.

**Figure 3 entropy-22-00946-f003:**
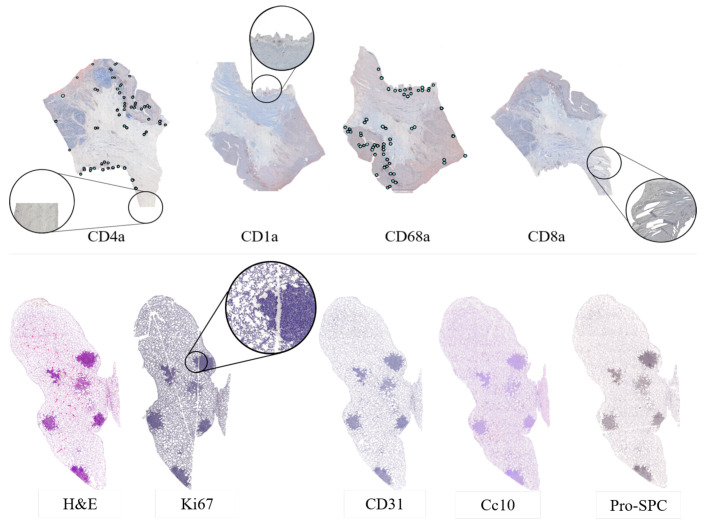
Examples of the images used. Top row: gastric adenocarcinoma tissue cuts from a patient, stained with the selected biomarkers. On top of the biopsies the manually annotated landmarks are mapped in blue. This clearly shows the typical challenges of performing registration on histopathology images with missing tissue (CD4a), rips (CD1A), rotations (CD68a), and folds in the tissue (CD8a). Bottom row: Lung Lobes tissue cuts from mouse, showing the complexity of the images, which are mostly mesh-like tissue with a few solid lesions.

**Figure 4 entropy-22-00946-f004:**
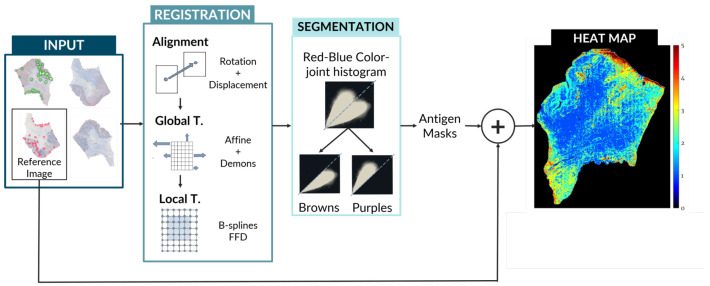
General view of the pipeline for our method. Steps of registration and segmentation up to the creation of the heat map of protein activity. Note: FFD, Free Form Deformation.

**Figure 5 entropy-22-00946-f005:**
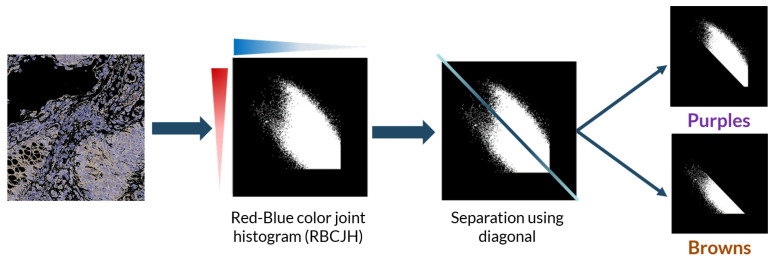
Basic explanation of the initial algorithm for positive antigen segmentation using the diagonal to divide the red-blue color joint histogram into the brown (positive antigen) and purple populations.

**Figure 6 entropy-22-00946-f006:**
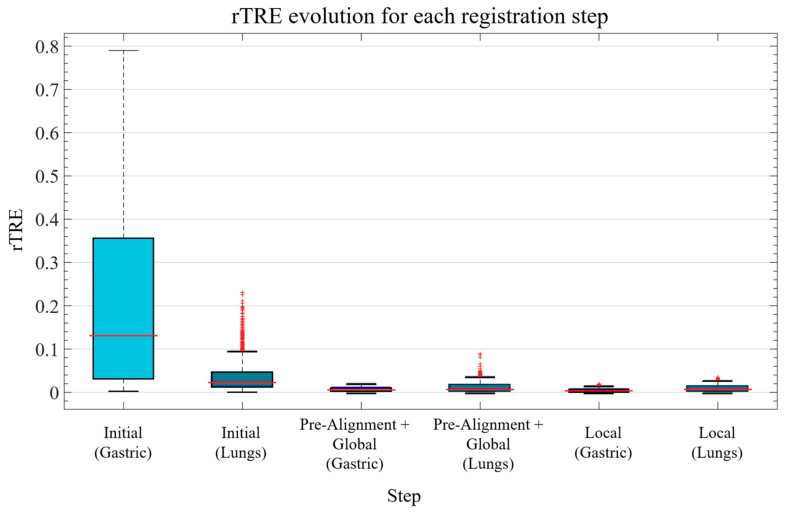
Boxplot of the rTRE measured for the Gastric and Lung Lobes datasets after each registration step.

**Figure 7 entropy-22-00946-f007:**
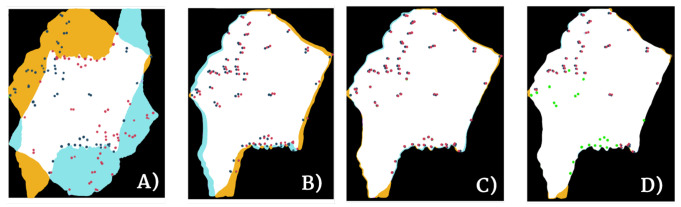
$T_{Alignment}$ and $T_{Global}$. The evolution of the landmarks is plotted on top of the overlap map. From left to right: (**A**) Initial position of biopsies; (**B**) Biopsies after alignment and rotation; (**C**) Biopsies after Affine transformation; (**D**) Biopsies after non-rigid Demons transformation. Red dots correspond to the landmarks of moving image (light blue). Blue dots correspond to landmarks of fixed image (yellow). Green dots represent complete overlap between landmarks.

**Figure 8 entropy-22-00946-f008:**
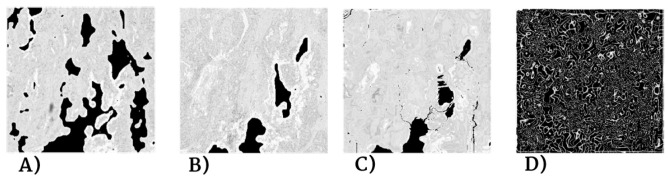
Example of how the local transformation model works on small details. Segment of 525 pixels × 525 pixels from original image: (**A**) Moving image, (**B**) Fixed image, (**C**) Transformed image, and (**D**) Final grid.

**Figure 9 entropy-22-00946-f009:**
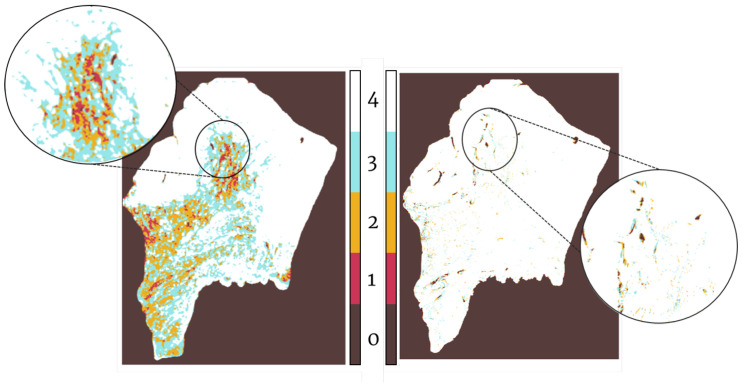
Degree of overlap between 4 biopsies. Left: Overlap after Global Registration. Right: Overlap after Local Registration. White represents complete overlap and red represents no overlap.

**Figure 10 entropy-22-00946-f010:**
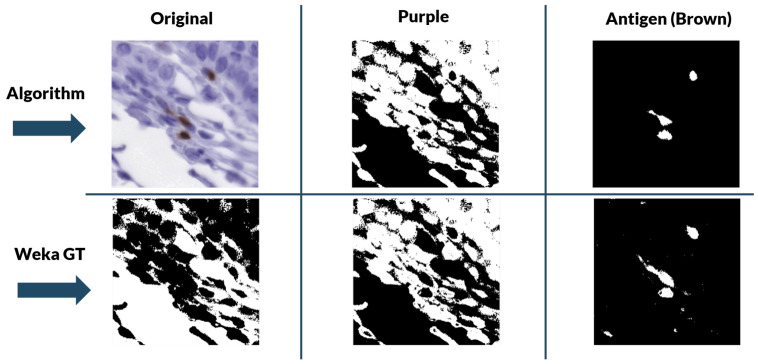
Color segmentation validation results (Background-Purple-Brown). In the top row are the results given by our color segmentation algorithm. In the bottom row are the WEKA results.

**Figure 11 entropy-22-00946-f011:**
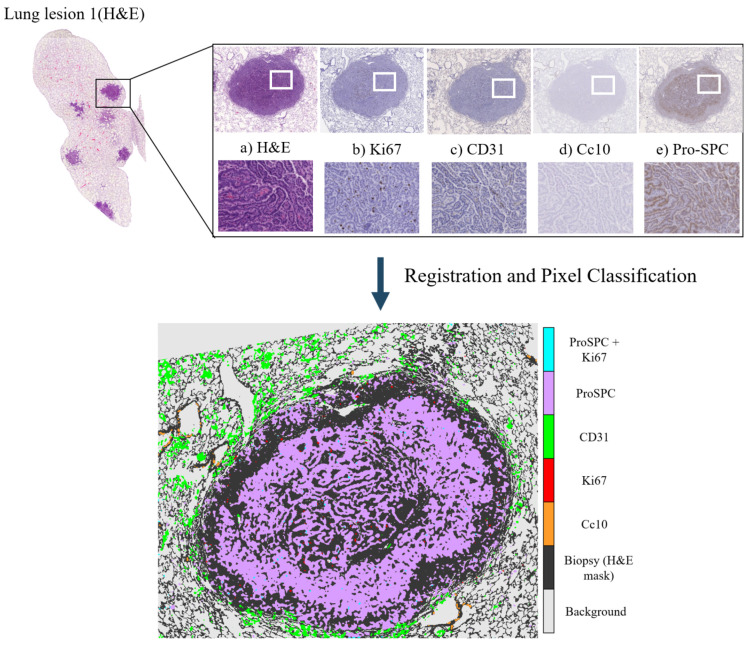
Creation of Tumor Heterogeneity Map measuring the distribution of the different stains over the lung biopsy. For clarity purposes, the map was created using just a small part of the Lung Lobes biopsy. At the top of the image there is a sample mosaic showing what the appearance of the different dyes confers to the slices: (**a**) Those slices with H&E staining nuclei are purple, the cytoplasm is pink, and vessels are red. Immunochemistry stains: Negative nuclei and cytoplasm are colored in dark and light blue, respectively. Positive content (nuclei/cytoplasm) is shown in dark brown. (**b**) Ki67 shows nuclei of growing dividing cells. (**c**) CD31: stains the endothelial cells that form blood vessels. (**d**) CC10: Clara cells found in the small airways (bronchioles). (**e**) ProSPC: type 2 pneumocytes forming the alveolar–capillary barrier. After registration and pixel classification of the resulting images, we obtain the Tumor Heterogeneity map showing the relations and possible colocalizations of the stains.

**Figure 12 entropy-22-00946-f012:**
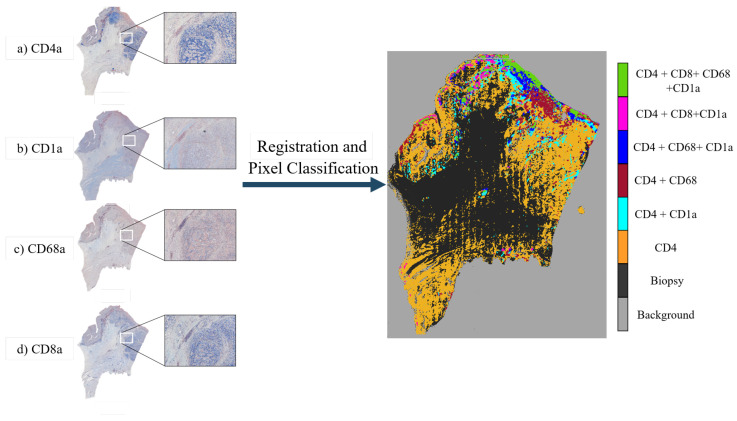
Creation of Tumor Heterogeneity Map measuring the distribution of the different stains over the gastric biopsy. At the left of the image there is a sample mosaic showing the appearance the different dyes confer the slices: (**a**) CD4a: Surface of helper T cells, (**b**) CD1a: Lipid antigen presenting molecule, and (**c**) CD68a: Histiocyte marker. Histiocyte: A stationary phagocytic cell present in
connective tissue. (**d**) CD8a: Cytotoxic-T cells

**Table 1 entropy-22-00946-t001:** Initial Parameters for Affine Registration.

Parameter	Value	
Gradient Magnitude Tolerance	$1.0\times 10^{-4}$
Minimum Step Length	$1.0\times 10^{-5}$
Maximum Step Length	$6.25\times 10^{-2}$
Maximum Number of Iterations	100
Relaxation Factor	0.8

**Table 2 entropy-22-00946-t002:** Table showing the landmark registration results of our algorithm used on the Gastric dataset compared to the main state-of-the-art methods (*).

Method	Average rTRE		Median rTRE		Max rTRE		Robustness	
	Average	Median	Average	Median	Average	Median	Average	Median
DROP *	0.158	0.002	0.162	0.002	0.317	0.012	0.862	1
ANTS *	0.176	0.059	0.171	0.057	0.322	0.086	0.656	0.721
RVSS *	0.067	0.002	0.067	0.002	0.124	0.006	0.846	1
BUNWARPj *	0.183	0.0313	0.18	0.031	0.323	0.06	0.747	0.721
ELASTIX *	0.162	0.003	0.164	0.002	0.311	0.0193	0.822	0.991
NIFTYREG *	0.186	0.036	0.181	0.039	0.326	0.064	0.686	0.633
OURS	0.006	0.007	0.006	0.007	0.01	0.009	0.92	1

**Table 3 entropy-22-00946-t003:** Table showing the landmark registration results of our algorithm used on the Lung Lobes dataset compared to the main state-of-the-art methods (*).

Method	Average rTRE		Median rTRE		Max rTRE		Robustness	
	Average	Median	Average	Median	Average	Median	Average	Median
DROP *	0.013	0.004	0.011	0.002	0.045	0.023	0.911	0.982
ANTS *	0.02	0.019	0.019	0.018	0.048	0.045	0.758	0.841
RVSS *	0.013	0.008	0.011	0.006	0.04	0.031	0.864	0.973
BUNWARPj *	0.026	0.026	0.025	0.025	0.057	0.06	0.649	0.638
ELASTIX *	0.004	0.004	0.003	0.003	0.022	0.018	0.972	0.984
NIFTYREG *	0.026	0.028	0.025	0.026	0.059	0.061	0.628	0.615
OURS	0.008	0.007	0.009	0.008	0.029	0.03	0.91	0.951

**Table 4 entropy-22-00946-t004:** Break down of the overlap after Global and Local Registration for the Gastric Dataset. Ideally, complete overlap percentage should be the highest; 1 means no overlap of the sections, 2 and 3 mean overlap of some but not all of the sections, and 4 means complete overlap of the 4 sections.

	AFTER GLOBAL REGISTRATION				AFTER LOCAL REGISTRATION			
Overlap	1	2	3	4	1	2	3	4
Median (%)	15.10	9.30	12.43	63.17	2.63	18.59	18.59	78.62
STD	6.31	4.20	4.20	9.64	1.41	28.51	28.51	8.64

**Table 5 entropy-22-00946-t005:** Break down of the overlap after Global and Local Registration for the Lung Lobes Dataset. In this case, there are 5 images per patient. Ideally, complete overlap percentage should be the highest; 1 means no overlap of the sections, 2–4 mean overlap of some but not all of the sections, and 5 means complete overlap of the 5 sections.

	AFTER GLOBAL REGISTRATION					AFTER LOCAL REGISTRATION				
Overlap	1	2	3	4	5	1	2	3	4	5
Median(%)	25.77	39.51	24.70	7.87	1.36	2.95	13.98	31.73	34.28	16.88
STD	0.22	0.20	0.05	0.02	0.33	0.01	0.02	0.06	0.02	0.13

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
