# Peer review of "Nonlinear Image Registration and Pixel Classification Pipeline for the Study of Tumor Heterogeneity Maps"

_entropy, 2020, doi:10.3390/e22090946_

Round 1

Reviewer 1 Report

This manuscript describes a methodology (a processing pipeline) for histopathology image registration that is largely automated (unsupervised). Image registration, especially across different image modalities (multiple IHC stains, or even IHC and H&E stains), is a very important problem in computational pathology that has not yet been resolved.

The present paper suggest a specific combination of multiple registration steps, and a colour segmentation step, to allow registration of multiple types of IHC stains. Although (most of) the individual components in the methodology have been reported previously, the authors put together a methodology/pipeline that offers automated (i.e. no human intervention needed) registration procedure that appears to show promising performance results. Effective (automated) registration of pathology images would be an important contribution to the field.

General comments (major):

- This manuscript has a severe problem in that it outlines a potentially interesting and relevant novel methodology, but then there is far from enough benchmarking results of the proposed method to allow reliable assessment of the merits of the method. For example, the method is only evaluated on one single  data set. That dataset is very, very small (few slides). The IHC stains are all against immuno targets (limited  diversity). The metrics used for evaluation, especially the most relevant metric, based on landmark registration, has reduced precision in the statistical metrics reported (only median). The manuscript and methodology proposed could have real impact in the community, but for this, proper benchmarking has to be performed and reported.

- The proposed method should be benchmarked against other state-of-the-art methods too, and on more than one data set. Maybe the authors could simply assess a few more of the ANHIR data sets, where presumably benchmarking results by other methods are already available.

- Tumour heterogeneity is important and a very interesting topic with high relevance in both tumour biology and clinical pathology. However, the analyses and discussion in this area is really limited in the manuscript. Could more detailed assessments, analyses, discussion be added?

- The dataset description lack detail, hard to understand without going to other sources, and this leads to problems in respect to evaluating the results presented.

- It is claimed that the proposed methodology is robust. However, robustness of the model cannot be assessed based on the present study, since only one dataset is evaluated, no sensitivity analysis (hyperparameters) are reported, and no other statistics/assessment related to robustness is reported. Generalisability across data sets are not assessed neither.

Specific comments (major):

- How does the method perform if some of the IHC stains have little or no signal (stain), it is expected that the uncertainty in registration would degrade in this case. Please make clear under which conditions the pipeline is expected to be effective, are there known limitations?

- What type of counterstain was used?

- Page 8: it seems like the parameter choices for the Local alignment could potentially lead to different performance: e.g. what is the sensitivity to choice of mesh spacing, basis function parameters (e.g. L247: 4um distance). Were such hyperparameters optimized? or how did you arrive at the used settings?. Consider to include information (supplementary information?) on sensitivity analyses, or hyperparameter optimization, for key tunable parameters in the registration pipeline, or explain why tuning is not required.

- Page 9: the choice of *median* rTRE may not be suitable to measure success in registration of pathology images, as this measure would disregard from failure in registration in potentially large areas of images, and still return “good” estimate of performance, due to the robustness properties of the median. You probably should also report broader distribution properties (boxplot or violinplot) of rTRE, rather than only the median. Alternatively, also report mean and 5%,25% 75% and 95% percentiles, for each slide (pairs) registered. This would provide at least a descriptive representation of the global performance across the range of landmarks. The Median rTRE in itself provide a too limited measure for registration performance, especially in situations where good performance throughout the whole image is required (typically the case for (clinical) pathology applications).

- It would be essential to in more detail describe the dataset used from the ANHIR challenge), it seems to be only 9 tumours? and 4 slides/sections from each patient (36 in total?). There is a confusion between biopsies and (consecutive) sections/slides from the same biopsy in the text (several places).

-L346, “shows the overlap for all biopsies” - do you mean all sections (presumably from the same biopsy?). Also, L341, “4 biopsies per patient were evaluated”, again, should it be 4 sections?

-Table 2. unclear description in table header, for example, what metrics is used here, which slides are used. Describe in the figure header too. What does the four categories 1-4 mean. Information is also not very clear in the results section referring to Table 2.

-Section 4.2.1 You refer to “overlap”, does this actually mean DICE coefficient? if so, write that out, e.g. DICE. Currently it is hard to know exactly what “overlap” means (or is it simply intersection?).

-Section 4.3 The heterogeneity map is simply generated as the sum of positive stain (binary) across four registered IHC (CDx) stains? If this is the case, it is misleading to use a continous heatmap, as your pixel level representation here is discrete 0,1,2,3, 4? I suggest you make a clear colour legend that illustrate the discrete nature of the colour.

- Please discuss the potential issue that the four IHC antibodies possibly would not stain the same areas of the slides, as they bind to different types of immune cells. I.e. biologically, it may not be expected that the same regions stain positive by different antibodies? How does this impact on the way you produce the heatmap, does it actually  visualize what you expect?

- p-14, line 380: the evidence for more accurate decisions can not be concluded from the present study. It might be the case, but the present study mainly presents a methodology, while no real-world testing is performed (in clinical setting) to assess diagnostic time, neither is any assessment of accuracy in clinical decision making performed. Remove statements, or at least make gentler claims.

- The methodology should probably be made available as software? Do you plant to publish the code - I sugget you do if possible.

Specific comments (minor):

- page 1, L18: spelling “8even”

Author Response

(see attached document)

Reviewer 2 Report

Dear authors,

I decided to create the review as I read the article in the form of comments to the individual lines. I am not an medical expert and yet I have some background in mathematics and image registration techniques as well as work with medical data. Please consider my contribution from this perspective.

To accept your manuscript I suggest the following changes:

Set up a public repository such as github, where you publish your source code. Provide examples how to use it.

Explain in the manuscript clearly the problem that needs to be adressed by the algorithms of registration and segmentation.

Improve results section. For example, put also the table with the actual numbers to the Figure 6. If its not possible to the manuscript, in terms of supplementary material or in terms of the data uploaded to github repository with the function to generate graph.

In terms of registration, explain why you decided to use "Gaussian multi-resolution pyramid" algorithm and reference it. Did you used another algorithms? If no, consider using standard OpenCV implementation of parametric registration https://docs.opencv.org/4.3.0/db/d61/group__reg.html based on http://www.cs.toronto.edu/~kyros/courses/2530/papers/Lecture-14/Szeliski2006.pdf

As OpenCV is de facto standard library for computer vision, it is worth trying to compare any image registration algorithm to a standard algorithm implemented there.

I would not call separating colors from single image "segmentation" as segmentation has different meaning for the image analysis that apparently does not match with what authors do.

How you finally would interpret the heterogenity map you generate? Is it something you would base some diagnostic decision on? Make an example.

In a present form I lean to reject the submission. However if all the issues I have raised are clarified, then I reconsider this and after a substantial amount of work on undestandability of the manuscript, clarifying input data sources and interpretation and clarifying application of presented algorithms it should be considered for publication as the method presents a great potential. It just needs to be explained better and anybody who reads the paper should be also able to use it by means of using or modifying author's reference implementation.

My particular comments follow:

Line 18: Typo 8even

Line 40: "They also estimate the size and reach of the primary tumor and whether it has to other parts of the body (staging)" ... its not clear how from single specimen pathologist can figure out whether "it has to other parts of the body".

Line 50: Image registration and image segmentation are a large subjects in computer science. I think that the example with apples is totally insufficient in describing what authors aim to achieve. In image registration, there is template image and image to be matched against the template. For example you do a CT scan pre and post operative and trying to align these images, where preoperative scan is the template. Here the role of the template and matching object was not clarified.

Figure 2: Needs further description in the caption.

Section 2: rather than extensive description of different algorithm I would like to see how they relate to the topic of the research. So I would like to see the aim what you want to register to what and then which structures you aim to segment. After this motivation is reasonable to discuss which algorithms could be used to achieve this.

Line 144: Were the patients informed about the inclusion to the study, were this study approved by some board? Or were the data publicly available? Comment on this.

Figure 3: Were the "landmarks" come from? Name algorithm.

157: Whole-Slide Image ... why you did not describe this in the introduction? As I understand it now, you do multiple slices from the same structure that are so thin that you can assume that they represent the same cut through the structure. You stain it and digitize it and then you are trying to match the structures to structures from different modalities? If this is a case, I don't understand why you do meaningless example with apples when there is real reasoning for image registration. Please describe the method!

What are landmarks in this context? 

Another question is how far apart are the slides. Are they oriented roughly the same? As in case of the CT registration where you see that the position of the relevant structures is roughly the same and you need to register details?

Or you need to do a potentially high dislocation matching such as half rotation and scaling? What motion model do you use for the registration. Is it rigid body motion? Euclidean motion, affine or even homeography? Or you expect that the motion model is something totally unpredictable and need nonglobal nonlinear motion models? 

167: I still do not understand landmarks. Are there from some algorithm such as SIFT? Or were they introduced by pathologists? What kind of annotation did pathologists provide to these landmarks and how many landmarks per slide were present?

178: It would be nice if you could publish all your source codes under some opensource licence such as GPL-3 and put it i.e. to github repository optimally including test data. Otherwise impact of your work will be limited as noone can test your actual algorithms.

179: As I have written earlier, you don't provide motion model, do not specify the size of the "sliding window" and do not explain the extent of the motion you expect from the structures to undergo. Please describe your algorithm precisely!

193: By which algorithm you do the preregistration? On whole images? What you select as template? On of the slices?

193,206: Is a global model what describes pre-alignment? Why you use two paragraphs then?

Have you been using the technique described in

https://doi.org/10.1016/j.neuroimage.2008.10.040

Why it is not referenced? Is it just a coincidence of the names of the technique? When you use this complex algorithm, did you implemented it on your own or is it a third party implementation. Comment on this.

229: Again the problem with specification of the problem. Could you in the introduction somehow clearly state how the images differ pre-global registration and pre-local registration? I appreciate the approach with the local registration, but it must be more clearly stated how the mask for the registration looks like, how the structures looks like and what is the benefit of the registration.

Equation 8: How to understand this? Element-wise or in terms of the sum? Is it matrix or scalar quantity? It seems that this metric does not take into account any changes in the image intensity between the samples. Comment on if this is not a problem and whether it would be an option to use something more involved such as ECC, see https://www.learnopencv.com/image-alignment-ecc-in-opencv-c-python/

Line 284: If the data are from the challenge, half of the data with landmarks. I suppose that second half has the landmarks hidden. Who then decides what was correct position of the marks in unmarked images? Comment on this.

Line 293: I am totally missing any background on what is color segmentation and what it is good for. Are these colors colors of different stains? Is this segmentation performed on registered data? Is it performed on single slice or multiple slices?

Moreover segmentation is normally refering to actual finding of some structure as a "brain tissue" or bone. There must be ground truth to compute something as Dice coefficient. And there is not described how the ground truth was formed.

304: typo algorithm

176 typo cores

Figure 5: Its not very illustrative, I don't understand the meaning of the arrows. Are you claiming that image on the left somehow transforms to the image on the right? Explain how!

321: What was ground truth? You have to have it to compute defined measures. Its not apparent.

Figure 7: I wonder why are the biopsies initially rotated 180 degrees? Is that normal situation?

Figure 8: What we see here? How to decode this picture?

Figure 9: What was the metric of overlap here? That is important question how is the "signal present" computed? Is there some threashold?

Table 2: What is the maximum of the overlap? Is it 100%? Or if there is signal in one image and no signal in another it is not "overlap" although there could be 100% alignment?

Author Response

(see attached document)

Round 2

Reviewer 1 Report

The manuscript has been improved in several ways in the revision, mainly by adding some missing information and further clarifying some parts.

- Recommendation: The manuscript is extremely long it its current form. Without doubt it would be possible to reduce the length substantially without loss of information. I suggest that the text is shortened /edited to be more succinct, if possible…

Major concern:

Table 2 appears has some severe problems in that all methods are not tested on the same data??? To me, the current table do not make sense (it is expected that the performance in different data sets will vary a lot, i.e. context dependent). as the main value of this table would be to enable comparison between previously available methods and your proposed method ( but your method is only tested on LL and G data, while other methods tested on some larger/different dataset). if I understand it correctly, your pipeline is not tested on the same data as the other methods - this makes the whole table useless at best, and misleading at worst. Unless I misunderstood? I think you ideally should try to fix this, then a direct comparison between registration methods will be possible. Second best would be to run some of the other methods on the same LL and G data (maybe not evan all of them?). I do not understand why you did this - are the results from the other methods lifted from previously reports, i.e. you could not implement the methods/evaluation for practical reasons on your side?

Author Response

In this document, we merged the letter to the reviewer as well as the old version of the manuscript with the changes marked in red. We had to do it in this manner because almost all of the changes were deletions in the text, which could not be properly shown in the new manuscript.

This manuscript is a resubmission of an earlier submission. The following is a list of the peer review reports and author responses from that submission.